# The COVID-19 Pandemic and Ophthalmic Care: A Qualitative Study of Patients with Neovascular Age-Related Macular Degeneration (nAMD)

**DOI:** 10.3390/ijerph19159488

**Published:** 2022-08-02

**Authors:** Seán R. O’Connor, Charlene Treanor, Elizabeth Ward, Robin A. Wickens, Abby O’Connell, Lucy A. Culliford, Chris A. Rogers, Eleanor A. Gidman, Tunde Peto, Paul C. Knox, Benjamin J. L. Burton, Andrew J. Lotery, Sobha Sivaprasad, Barnaby C. Reeves, Ruth E. Hogg, Michael Donnelly

**Affiliations:** 1School of Psychology, Queen’s University of Belfast, Belfast BT7 1NN, UK; 2Centre for Public Health, Queen’s University of Belfast, Belfast BT12 6BA, UK; c.treanor@qub.ac.uk (C.T.); t.peto@qub.ac.uk (T.P.); r.e.hogg@qub.ac.uk (R.E.H.); michael.donnelly@qub.ac.uk (M.D.); 3Bristol Trials Centre (CTEU), University of Bristol, Bristol Royal Infirmary, Bristol BS2 8HW, UK; elizabeth.ward5@uhbw.nhs.uk (E.W.); r.a.wickens@soton.ac.uk (R.A.W.); lucy.culliford@bristol.ac.uk (L.A.C.); chris.rogers@bristol.ac.uk (C.A.R.); eleanor.gidman@bristol.ac.uk (E.A.G.); barney.reeves@bristol.ac.uk (B.C.R.); 4Southampton Clinical Trials Unit, University of Southampton, University Road, Southampton SO17 1BJ, UK; 5Exeter Clinical Trials Unit (EXECTU), St. Lukes Campus, University of Exeter, Exeter EX1 2LT, UK; a.j.oconnell@exeter.ac.uk; 6Department of Eye and Vision Science, University of Liverpool, Liverpool L7 8TX, UK; pcknox@liverpool.ac.uk; 7James Paget University Hospitals NHS Foundation Trust, Norfolk NR31 6LA, UK; ben.burton@jpaget.nhs.uk; 8Department of Clinical and Experimental Sciences, Faculty of Medicine, University of Southampton, Southampton SO16 6YD, UK; a.j.lotery@soton.ac.uk; 9NIHR Moorfields Biomedical Research Centre, Moorfields Eye Hospital NHS Foundation Trust, London EC1V 2PD, UK; sobha.sivaprasad@nhs.net

**Keywords:** COVID-19, patient perspective, ophthalmic care, qualitative methods

## Abstract

Concerns have been expressed about the relationship between reduced levels of health care utilisation and the COVID-19 pandemic. This study aimed to elicit and explore the views of patients with neovascular age-related macular degeneration (nAMD) regarding the COVID-19 pandemic and their ophthalmic care. Semi-structured telephone interviews were conducted with thirty-five patients with nAMD taking part in a larger diagnostic accuracy study of home-monitoring tests. Participants were recruited using maximum variation sampling to capture a range of key characteristics including age, gender and time since initial treatment. Transcribed interview data were analysed using a deductive and inductive thematic approach. Three themes emerged from the analysis: i. access to eye clinic care. ii. COVID-19-mitigating factors and care delivery and iii. social and personal circumstances. Participants reported anxieties about cancelled or delayed appointments, limited communication from clinic-based services about appointments, and the impact of this on their ongoing care. Despite these concerns, there was apprehension about attending appointments due to infection risk and a perception that nAMD patients are a ‘high risk’ group. Views of those who attended clinics during the study period were, however, positive, with social distancing and infection control measures providing reassurance. These findings contribute to our understanding about experiences of patients with nAMD during the COVID-19 pandemic and may have potential implications for future planning of care services in similar circumstances. Innovative approaches may be required to address issues related to access to care, including concerns about delayed or cancelled appointments.

## 1. Introduction

The World Health Organization (WHO) formally declared Coronavirus disease 2019 (COVID-19) as a pandemic on 11 March 2020 [1]. COVID-19 is an infectious acute respiratory disease caused by a novel coronavirus (SARS-CoV-2) [2]. Older people and those with underlying health conditions are at increased risk of developing serious illness that may have significant longer-term health effects [3]. Ophthalmology clinics may be impacted by COVID-19, as clinicians often perform assessments, examinations and deliver treatments to patients in close proximity. Methods of remote care in place of clinic-delivered treatment may therefore be less viable in comparison to other areas of clinical practice [4]. As a result of this and other structural factors, many clinic services stopped or substantially reduced the usual schedule of clinical appointments and procedures. This decrease in service provision subsequently led to indirect health effects during the COVID-19 pandemic [5]. Indirect effects tended to be associated with increased patient morbidity because of restricted preventative care services, diagnostic delays and reduced intervention delivery [6]. More broadly, national-level and localised lockdowns, and social distancing measures, contributed further in terms of restricted patient access to ongoing care. Furthermore, public information and advice that was designed to mitigate the effects of COVID-19 transmission appeared to promote a perception that attending a hospital appointment or an unscheduled clinic visit could increase the risk of contracting the virus [7]. Age-related macular degeneration (nAMD) is a chronic, progressive condition and the commonest cause of vision loss in older adults [8]. Global prevalence of AMD is predicted to increase from 196 million in 2020, to 288 million in 2040 [9]. Neovascular AMD (nAMD) is a form of late AMD and is often associated with irreversible visual loss. It accounts for around 90% of cases of severe sight impairment [10,11]. Ongoing surveillance is necessary to manage disease activity since nAMD can recur following periods of treatment which is inconvenient for patients and costly for the health service [12]. Therefore, it is important to examine changes to clinic-based services to improve efficiency in relation to patient outcomes. This qualitative study elicited and explored the views of patients with nAMD regarding the COVID-19 pandemic and changes to their ophthalmic care.

## 2. Materials and Methods

Qualitative methods were used to explore patients’ responses, views and experiences, and to examine variations in personal contexts [13]. Participants were interviewed at least one month after the COVID-19 pandemic was declared by the WHO [1] and once public health measures were in place in the UK [14]. The study followed the consolidated criteria for reporting qualitative research (COREQ) [15]. Ethical approval was acquired from the National Research Ethics Service (IRAS ref: 232,253 REC ref: 17/NI/0235).

### 2.1. Participants

Remote, semi-structured interviews were conducted between 29 April and 14 September 2020, covering the period during the first national lockdown in March 2020 and the easing of restrictions in July 2020. The study included a subset of nAMD patients who were taking part in a larger diagnostic accuracy study of home-monitoring tests (MONARCH) [16]. Participants were recruited from four sites within the UK. Maximum variation sampling was used to ensure that a range of perspectives were captured in relation to age category (young–old 50–69 years and older-old 70+ years), gender, laterality of nAMD (unilateral and bilateral) and time since first treatment (6–17 months, 18–29 months and 30–41 months). Participants were given a minimum of one week to consider the study information and discuss it with family members before agreeing to take part. Informed consent to participate in the study was obtained verbally prior to interviews and following a full explanation of study procedures.

### 2.2. Data Collection

All participants were given the option of completing the semi-structured interview using video-conferencing software, or via telephone. The interview schedule (see Appendix A) was developed based on the experience of the research team and was informed by relevant theoretical models including the Theoretical Framework of Acceptability [17]. Members of the team who collected and analysed the data (CT, SOC, MD) had extensive experience in the application of qualitative methods in healthcare research. No participant was known to the researchers who conducted the interviews.

### 2.3. Data Analysis

Interviews were audio-recorded and transcribed. A directed content analysis approach based on deductive and inductive coding was used [18]. Coding underwent iterative development as individual transcripts were reviewed and re-reviewed during data familiarisation (CT, SOC, MD) (See Appendix A). Following line-by-line coding of each transcript (CT, SOC), findings related to views on usual care, the impact of COVID-19 on care, and views about the COVID-19 pandemic in general were summarised. Transcripts were cross-coded and discussed to ensure rigour and reflexivity. Related codes were clustered and grouped into themes that were reviewed and refined to ensure coherence. NVivo version 12 was used to manage data and facilitate the analysis process. This process, in summary, included the following stages: i. independent transcription, ii. data familiarisation, iii. independent coding, iv. development of an analytical framework, v. indexing, vi, charting and vii. interpreting data.

## 3. Results

Two participants who were approached declined to take part. A total of 35 interviews were completed. Interviews took place in the context of organisational and structural changes to clinic services including the cancelation of routine appointments and prioritisation of urgent care. In some cases, virtual methods were used to ‘triage’ patients and identify if there was a need to attend an urgent appointment. All participants opted to complete the remote interviews by telephone. The demographics of participants are shown in Table 1. The majority of participants were female (69%) with a mean age of 77 years. Interviews lasted an average of 48 min (range: 39 to 78 min). With respect to their experiences of COVID-19, one participant reported having had a negative test for COVID-19 and two reported that they had experienced suspected coronavirus related symptoms previously. Three participants were advised to self-isolate due to close contact with a person diagnosed with COVID-19. No participants were diagnosed with COVID-19 at the time of their interview. Three overarching themes emerged from the analysis and revolved around: i. access to eye clinic care, ii. COVID-19-mitigating factors and care delivery, and iii. social and personal circumstances. Aspects of the three themes overlapped. Each theme is presented in the section below. Selected illustrative quotes from participants regarding their views about access to care, and the factors that were used to mitigate changes to care delivery are presented in Table 2. Key quotes related to views about the influence of COVID-19 on social and personal circumstances are presented in Table 3.

### 3.1. Theme i. Access to Eye Clinic Care

Participants became concerned when their access to eye clinic care was restricted as a result of changes to services during the COVID-19 pandemic and worried that clinics would remain closed during the pandemic—the vast majority had expected or previously arranged appointments cancelled by the time of their qualitative research interview. Greater concerns about not attending an appointment were apparent among patients who described having attended clinic appointments at regular intervals (e.g., every four to six weeks) before the pandemic and who were, therefore, expecting to attend an appointment when changes to clinical services were implemented. More specifically, patients reported fears about potential worsening of symptoms or deterioration in their vision and uncertainty about what to do in this event. Some participants described how they felt that deterioration had occurred. This was reported typically by patients who had continued to receive treatment during the pandemic, and was therefore attributed by these patients to normal disease progression and not to the impact of the pandemic on services. It was also highlighted that participants missed the face-to-face contact and reassurance that clinic visits provided, but the need for organisational changes due to infection risk during the pandemic, resulting in limited access to services, was acknowledged. It was also assumed that these changes were a result of clinical staff being temporarily redirected to other areas, as this was something that participants described noticing in other areas of the health service.

The importance and value of effective communication was emphasised particularly in relation to information for patients about delayed appointments and their rescheduling. Actual experiences of communication from clinics were mixed. Some participants, who had not been contacted about appointments, were unsure why this was the case, and were not aware that prioritisation of urgent care was in place. In some cases, participants had initiated contact with services and enquired about the rescheduling of appointments, though others did not feel able or willing to contact their clinic. Those participants who contacted services tended to be younger and female.

Although no participants had taken part in any remotely delivered or virtual eye clinic appointments, some had done so in other care contexts, primarily as part of primary care. It was highlighted that these methods could be a possible, albeit temporary solution which might be used by services to enquire about and ‘assess’ any changes in vision. Other ‘models’ of care were suggested by participants to ensure access to ongoing care, included the use of community-based optometry services for assessing nAMD progression. These treatment delivery arrangements were also viewed as possible ways in which to relieve the burden on hospital eye services beyond the COVID-19 pandemic.

### 3.2. Theme ii. COVID-19-Mitigating Factors and Care Delivery

While there were fears around deterioration in vision, or worsening of other symptoms, participants were often apprehensive about attending hospital appointments as part of their eye care, or for management of co-morbidities or other existing conditions. This was due to the perceived risk of COVID-19 infection, and the risk of severe illness if infected due to the older age of nAMD patients. Despite this, the opinions of the small number of patients (approximately one third) who did attend eye clinic appointments during the study period were typically very positive. Participants highlighted the professionalism of clinic staff, and felt reassured that mitigating factors including social distancing and infection control measures were being strictly adhered to. Appointments were compared to the experience of visiting clinics prior to the pandemic. Previous visits were often described as being lengthy, involving long waiting times and held in busy and sometimes crowded environments. As a result of COVID-19-mitigating factors, appointments during the pandemic were seen as involving markedly shorter waiting times, clear social distancing and with appropriate use of personal protective equipment. One negative aspect for participants was that partners or relatives who usually attended appointments with patients were unable to do so and therefore could not provide support during visits, or if usually provided, practical assistance such as providing transport to clinic appointments. An additional concern raised was that participants worried about the risk of partners or family members contracting coronavirus infection after they had visited hospital sites.

### 3.3. Social and Personal Circumstances

Participants reported experiencing a sense of social isolation, as well as generalised anxiety related to the impact of the pandemic, and ambiguity about the long-term effects on their care and overall quality of life. Many described how the pandemic had a significant effect on their routines and felt that lockdowns and social distancing requirements, and the need to self-isolate, had a major influence on their social interactions and in some cases, employment status. In a few cases, participants described having to move temporarily and live with other family members. It was felt by some that measures were applied without sufficient information being provided around how long they might be in place, and for these participants, this increased their sense of uncertainty. Despite feeling that health information was conflicting, participants also felt strongly about adhering to health measures. This was most apparent in those participants with comorbidities, including cancers, and respiratory conditions, who stated that they felt at increased risk. There was a clear understanding around the reasons for health measures, even if it resulted in changes to routines, and views were strongly negative towards those who do not adhere to public health measures.

## 4. Discussion

In the present study, evidence is provided which can contribute towards an improved understanding of the impact of the COVID-19 pandemic on patients with nAMD. The importance of conducting qualitative research during the COVID-19 pandemic has been highlighted [19]. Qualitative methods are essential to provide in-depth exploration of patient perspectives and can also be used alongside longitudinal or retrospective study designs to assess the effect of COVID-19 on patient outcomes.

Three themes emerged from the analysis. These related to access to care, the effect of mitigating factors on care delivery, and the influence of patients’ social and personal circumstances. Concerns were reported about limited access to care, and missed or delayed eye clinic appointments, but there was a common understanding around the reasons for the organisational changes to services because of the pandemic. Participants reported experiencing a sense of social isolation, as well as generalised anxiety related to the impact of the pandemic, and uncertainty about the long-term effects on their care and overall quality of life. While there were fears around deterioration in vision, or worsening of other symptoms, participants were also often apprehensive about attending hospital appointments as part of their eye care, or for management of co-morbidities or other existing conditions. This apprehension was related to perceived infection risks and a view that nAMD patients as an older population were therefore a ‘higher risk’ group. An additional concern raised was that participants worried about the risk to partners or family members of contracting coronavirus infection after they had visited hospital sites. Participants also highlighted that support from family members in terms of attending clinic appointments alongside patients was affected by COVID-mitigating changes to clinics. Despite these observations, the opinions of patients who attended clinical appointments during the study period were positive. Participants highlighted the professionalism of clinic staff, and their strict adherence to measures to mitigate the risk of infection, including social distancing and personal protective equipment. In an earlier study [20], information and support, as well as additional factors which influence service delivery, such as appointment and waiting times, were highlighted as potential targets to improve patients’ experience of being assessed for and receiving treatment for AMD. In the present study, the importance and value of effective communication with services was also emphasised. This was seen as important particularly in relation to patients being provided with information on delayed appointments, and when they might be rescheduled. It was interesting that only some participants felt it was important to contact services to enquire about future appointments and others did not. This may have been related to the stage of nAMD, e.g., whether patients were receiving active treatment before the pandemic or surveillance only. These different reactions to their circumstances would merit more detailed examination.

Our findings are broadly reflective of those reported in other qualitative studies examining the impact of COVID-19 in patient populations. Concerns about restricted access to care and social isolation as a result of the pandemic have been reported in different groups, including those with chronic pain disorders [21] diabetes [22] and obesity [23]. Studies have confirmed that the pandemic has been associated with reductions in routine assessments and treatment in various clinical areas, including ophthalmology [5,6,24]. Findings from quantitative survey studies have also highlighted the possible influence of COVID-19 related fear on care continuity in ophthalmology care contexts. In one study [25], around half of participants with AMD or diabetic retinopathy were at least moderately concerned about vision loss due to missed or delayed treatment; and concern relating to COVID-19 exposure during appointments was a factor associated with higher loss to follow-up (odds ratio [OR], 3.9; 95% CI, 1.8–8.4). A further study [26] found that a lower number of participants (16%) were fearful of visual loss due to difficulties in maintaining regular follow-ups. However, female participants were more likely to postpone appointments, potentially due to higher levels of COVID-19 related anxiety.

Going forward, the need for reorganisation of services to reduce the effects of service change on patient outcomes is acknowledged [27]. For example, it is recognised that it is likely that the need for remotely delivered care will continue to increase [28,29]. Other models of care may also be required, including increased use of community-based optometry services for managing nAMD, to relieve burden on hospital eye service. However, potential barriers to use of such services have been previously identified, including concerns about potential delays in referrals for intervention when it is required [30].

A potential limitation of the study is that use of remote telephone interviews may provide different information than would be gathered using face-to-face interviews [31,32]. This method was, however, precluded because of the COVID-19 pandemic and associated social distancing measures. The study was also based on interviews conducted with participants taking part in an ongoing diagnostic accuracy study. Questions around the impact of COVID-19 were therefore not the only focus of the interviews and the responses may have been less in depth than if they related only to COVID-19. While participants were recruited to the study using maximum variation sampling methods (to ensure a balance of important patient characteristics), the sample may also not be reflective of all patients with nAMD. The number of male participants recruited to the study was also relatively low. Participants were also recruited from sites within the UK and findings may not be applicable to other healthcare systems. Another limitation is that findings could have been influenced by the different phases of the pandemic at which the interviews were conducted.

## 5. Conclusions

This study is one of only a few studies which report the perspectives of nAMD patients regarding the impact of the COVID-19 pandemic on their care. Three themes emerged from the analysis related to concerns about access to care, the effect of mitigating factors on care delivery, and the influence of patients’ individual circumstances. The most significant factor was the impact on access to care. Participants emphasised the importance and value of effective communication by services to address these concerns. Participants also suggested how alternative models of care could play a part in managing issues around access to care. These alternatives included effective and easily implementable remote methods of delivery and increased use of other models of care such as optometry services to reduce the burden on ophthalmology services at times of strain on hospital-based systems. There is a need to consider strategies that would reduce or avoid the negative influence of restrictions (that need to be implemented to mitigate risk) on the support provided by family members to patients to attend clinic appointments.

In summary, these findings could be used to understand the experiences of patients with nAMD during the ongoing COVID-19 pandemic, and could have implications for future planning of care services in the event of subsequent waves, or future pandemics. Innovative approaches may be required to address the issues raised related to patients’ concerns about ensuring adequate access to care. Findings also highlight the importance of ensuring adequate referral pathways are in place to provide patient reassurance, and to facilitate rapid referral in the case of suspected deterioration in vision [33,34]. Consideration should also be given to supporting patients to manage social isolation and anxiety in vulnerable patient groups, including those in older populations, and those who have existing co-morbidities or chronic health-related conditions such as nAMD. Further studies examining the indirect health effects of the COVID-19 pandemic in ophthalmology are also essential to improve understanding of its impact on longer-term patient and service level outcomes.

## Figures and Tables

**Table 1 ijerph-19-09488-t001:** Demographic characteristics of participants.

Baseline Characteristics		Sample (n = 35)	
Sex	Male	n	%
	Female	11	31.4
Age	Mean (SD) years	24	68.6
Visual acuity *	Mean (SD) LogMAR	77.4 (8.4)	-

* For patients with two involved eyes, better seeing eye is used.

**Table 2 ijerph-19-09488-t002:** Selected quotes related to views on the impact of COVID-19 on access to care and the effect of mitigating factors on care delivery.

Theme: Impact on Access to Care
‘… the only thing is, it has been a long, long time since they have called me in. I understand the reason, but I am concerned because the second eye is getting worse, and I am hoping that they will call me in soon.’ (Male, Age range: 71–75 years, #30)
‘… I know I’m supposed to be there every four to six weeks, but it’s a lot longer than that this time. But, I understand, they can’t help it.’ (Male, Age range: 71–75 years, #14)
‘… yes, I notice things are getting worse. Originally, it was just my left eye giving the trouble, and it has got worse. I am finding it harder to focus. The last time I was there, the right eye was starting to go, so I had an injection in it too. I am concerned that it is going to get worse the longer this goes on.’ (Female, Age range: 61–65 years, #19)
‘… so COVID already had come in, so there was the distancing, things like that there. But, the reason this time for the delay, was that they had to put in more measures, and I accept that. But, they could have let me know.’ (Male, Age range: 71–75 years, #20)
‘… it was the {date} I got my first injection in the left eye and then they brought me back on the {date} Then, it was the start of August, so I got three in a row. I haven’t heard anything since and to be honest, I ended up ringing them because I was so anxious as both eyes are now wet. I rang, and they did ring me back and sort out an appointment pretty quick.’ (Female, Age range: 61–70 years, #32)
‘… I’m supposed to get injections every six to eight weeks. But with COVID and what have you its overdue now and I don’t know when I‘m going to get the next one. It does feel like things improve or are at least stabilized with the injections.’ (Female, Age range: 61–70, #24)
‘… in the meantime, I have had a visit to the opticians, and he was happy my eyesight had been much the same as it had been the previous visit to him a year ago. That was {date} I went there, so it’s a bit reassuring to know that.’ (Female, Age range: 71–75 years, #09)
**Theme: Effect of COVID-19-Mitigating Factors on Care Delivery**
‘… I don’t know what’s going to happen now in the next period of time, because it’s very difficult in there to separate people, they are all crammed into a small unit.’ (Female, Age range: 66–70 years, #33)
‘… well, I have been fine with my appointments, everything has went quite straightforward. The last one after COVID had started, it was all social spacing, and that was excellent at the hospital.’ (Female, Age range: 61–65 years, #07)
‘… I would have preferred to stay at home than be out. I suffer from COPD and going to a place where there would be a lot of people, unless it is really necessary, I would have preferred not to have gone.’ (Male, Age range: 66–70 years, #28)
‘… It is no good saying, oh it is not going to affect me, I am ok. I was worried because I am {age}, which puts me in the top band. Then, to be told in the last month I also have cancer in my lung, makes me very, what would you say, nervous about going out.’ (Male, Age range: 71–75 years, #14)
‘… everybody at the clinic was, honestly they were so good and so reassuring. No matter how many times you go to get the injection, you just are really, really nervous. The waiting time, we were usually there for two hours but then when COVID kicked in, it was actually a lot quicker because they can’t have as many in the clinic.’ (Female, Age range: 71–75 years, #11)
‘… they got me an appointment within a week and confirmed what I thought, that it had turned wet. Obviously, then I got the injection. With the whole COVID thing, my left eye, that happened in the middle of it all, but the clinic were very good.’ (Female, Age range: 66–70 years, #18)
‘… yeah, I have been lucky enough because I had one at the end of April which was cancelled, so that was it, but in fact I got an appointment again in May to come. I found it was fine, it was very different, very few people there and before you get through the door, you are tested. Nobody is allowed to come with you, you have to go on your own. You have your temperature taken and if that is ok then you go through the whole routine, your hands sanitised and everybody there is covered in masks and visors, and aprons. So, it is very safe.’ (Female, Age range: 71–75 years, #04)
‘… I have had two since then, although the last time I did get my eyes scanned, so maybe that was because things are so different now, there are less staff around certainly.’ (Male, Age range: 71–75 years, #22)

**Table 3 ijerph-19-09488-t003:** Selected quotes related to views on the influence of COVID-19 on social and personal circumstances.

Theme: Influence of Social and Personal Circumstances
‘… so for six weeks, when they shut down, it all happened very quickly here, in my family they live down in [town] and it was all a bit scary at the beginning. They insisted on me going down there, I‘ve stayed with my son since then.’ (Male, Age range: 71–75 years, #14)
‘… I suppose I do feel more depressed now with this happening. Every so often its just feels difficult, you know, you just really really need to go and get out of the house.’ (Female, Age range: 66–70 years, #18)
‘… I have been getting worried, {partner name} looks like his job is going to end, that means we won’t have as much support any more.’ (Female, Age range: 66–70 years, #03)
‘… we are all paying attention to the precautions and the rules and I do take them. It can be a bit scary at times but I understand why it’s needed right now.’ (Male, Age range: 66–70 years, #12)
‘… I would prefer to stay at home than be out. I suffer from COPD and going to a place where there would be a lot of people, unless it is really necessary, I would have preferred not to have gone.’ (Male, Age range: 66–70 years, #28)

## Data Availability

Study data are available from the corresponding author upon request.

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
