# Peer review of "The COVID-19 Pandemic and Ophthalmic Care: A Qualitative Study of Patients with Neovascular Age-Related Macular Degeneration (nAMD)"

_ijerph, 2022, doi:10.3390/ijerph19159488_

Round 1
Reviewer 1 Report
ABSTRACT
Very well written and clear. The only suggestion is that a sample of 35 may be insufficient to suggest broad recommendations.
INTRODUCTION
Line 62: suggest replacing “influence” with “promote”
Line 64: it would be informative that. Given that AMD is a very well known entity, it be specified that nAMD includes 15% of AMD cases and has much more risk for visual loss.
Otherwise, excellent!
MATERIALS AND METHODS
2.1 Participants: please add that verbal informed consent was approved by the IRB
Otherwise, the Methods are clear and complete.
RESULTS
Initial paragraph clear and complete.
Line 43 is capitalized but I believe an extension of line 42.
Lines 143 to 169 are mainly qualitative but quite informative indicating that patients with potential visual loss felt constrained in access to periodic testing and quantification of visual loss.
There seems to be a gap in completion of line 175 but this reviewer may have missed. I suggest clearer introduction/explanation in the text of the various quotes reported by patients which are very informative. The description of those patients who attended appointments is informative but suggests that such appointments could be accompanied by telephonic by these patients’ family members who normally attend the appointments with the patients.
DISCUSSION
Very well written. Perhaps suggest that the inclusion of more definitive home monitoring techniques could mitigate issues identified in this unique groups of subjects.
CONCLUSIONS
All points are well stated but again suggest that the inclusion of more definitive home monitoring, inclusion of family members in direct or telephonic interviews and stressing contacts for immediate intervention in the event of substantive changes in vision.
Author Response
ijerph-1798250 |
|
The COVID-19 pandemic and ophthalmic care: a qualitative study of patients with neovascular age-related macular degeneration (nAMD) |
|
Reviewer comments |
Authors response |
Reviewer #1 |
|
ABSTRACT: …a sample of 35 may be insufficient to suggest broad recommendations. |
Thank you for your comment about this important point. The sample met criteria (e.g. data saturation) that indicated data adequacy for a qualitative study and we have taken care to ensure that conclusions based on the findings reported in the paper were expressed tentatively - see amended (Line 39). The purpose of this qualitative analysis like many qualitative studies was to present the perspectives of patients rather than to make recommendations. “These findings contribute to our understanding about experiences of patients with nAMD during the COVID-19 pandemic and may have potential implications for future planning of care services. Innovative approaches may be required to address issues related to access to care, including concerns about delayed or cancelled appointments”.
|
INTRODUCTION: Line 62: suggest replacing “influence” with “promote” |
Thank you. We have made this change at line 63.
|
INTRODUCTION: Line 64: it would be informative that. Given that AMD is a very well known entity, it be specified that nAMD includes 15% of AMD cases and has much more risk for visual loss. |
Thank you for your comment. We have amended the text at line 65 accordingly.
“Age-related macular degeneration (nAMD) is a chronic, progressive condition and the commonest cause of vision loss in older adults[8]. Global prevalence of AMD is predicted to increase from 196 million in 2020, to 288 million in 2040[9]. Neovascular AMD (nAMD) is a form of late AMD and is often associated with irreversible visual loss. It accounts for around 90% of cases of severe sight impairment[10,11]”.
Added references: Keenan TDL, Cukras CA, Chew EY. Age-Related Macular Degeneration: Epidemiology and Clinical Aspects. Adv Exp Med Biol. 2021;1256:1-31.
Shao J, Choudhary MM, Schachat AP. Neovascular Age-Related Macular Degeneration. Dev Ophthalmol. 2016;55:125-36.
|
MATERIALS AND METHODS: 2.1 Participants: please add that verbal informed consent was approved by the IRB |
Thank you for your comment. Details of ethical approval covering consent procedures are mentioned in the preceding paragraph at Line 80.
“Ethical approval was acquired from the National Research Ethics Service (IRAS ref: 232253 REC ref: 17/NI/0235)”. |
RESULTS: Line 43 is capitalized but I believe an extension of line 42. |
Thank you. This line has been amended. |
RESULTS: There seems to be a gap in completion of line 175 but this reviewer may have missed. |
Thank you. This line has been amended. |
I suggest clearer introduction/explanation in the text of the various quotes reported by patients which are very informative.
|
Thank you for your comment. We have amended the text at line 132 accordingly, to introduce the quotes.
“Selected illustrative quotes from participants regarding their views about access to care, and the factors that were used to mitigate changes to care delivery are presented in Table 2. Key quotes related to views about the influence of COVID-19 on social and personal circumstances are presented in Table 3”.
|
The description of those patients who attended appointments is informative but suggests that such appointments could be accompanied by telephonic by these patients’ family members who normally attend the appointments with the patients. |
Thank you. We have amended the section in the discussion at line 241 to emphasis this important point.
“Participants also highlighted that support from family members in terms of attending clinic appointments alongside patients was affected by COVID-mitigating changes to clinics. Despite these observations, the opinions of patients who attended clinical appointments during the study period were positive”.
We have also added a line in the conclusion to address this point further (Line 298)
“Strategies should also be considered to ensure support provided by family members who usually attend clinic appointments alongside patients, is not influenced negatively by restrictions put in place to mitigate transmission risk”.
|
DISCUSSION: …suggest that the inclusion of more definitive home monitoring techniques could mitigate issues identified in this unique groups of subjects. |
Thank you for your comment. We have amended the text at line 295 to emphasise this important point.
“This included effective and easily implementable remote methods of delivery”.
|
CONCLUSIONS: …suggest that the inclusion of more definitive home monitoring, inclusion of family members in direct or telephonic interviews and stressing contacts for immediate intervention in the event of substantive changes in vision. |
Thank you. We have added the following text at lines 288 and 300 to address this.
“Findings also highlight the importance of ensuring adequate referral pathways are in place to provide patient reassurance, and to facilitate rapid referral in the case of suspected deterioration in vision”.
“Participants also suggested how alternative models of care could play a part in managing issues around access to care. These alternatives included effective and easily implementable remote methods of delivery and increased use of other models of care such as optometry services to reduce the burden on ophthalmology services at times of strain on hospital based systems. There is a need to consider strategies that would reduce or avoid the negative influence of restrictions (that need to be implemented to mitigate risk) on the support provided by family members to patients to attend clinic appointments”.
|

Reviewer 2 Report
Study is useful in its exploration of patient perspectives but is not technically novel as stated by the authors. Study exploring patient perspectives more specifically related to fear and anxiety already exists in the literature and was not cited as one of the references for this article.
Author Response
ijerph-1798250 |
|
The COVID-19 pandemic and ophthalmic care: a qualitative study of patients with neovascular age-related macular degeneration (nAMD) |
|
Reviewer comments |
Authors response |
Reviewer #2 |
|
Study is useful in its exploration of patient perspectives but is not technically novel as stated by the authors. |
Thank you. We have amended the text at line 282 accordingly.
“This study is one of only a few studies which report the perspectives of nAMD patients regarding the impact of the COVID-19 pandemic on their care”.
|
Study exploring patient perspectives more specifically related to fear and anxiety already exists in the literature and was not cited as one of the references for this article. |
Thank you for your comment. We have discussed patient concerns relating to anxieties about reductions in access to care (Line 262) across different conditions and included an additional reference related to this point.
“Our findings are broadly reflective of those reported in other studies examining the impact of COVID-19 in patient populations. Concerns about restricted access to care and social isolation as a result of the pandemic have been reported in different groups, including those with chronic pain disorders[21] diabetes[22] and obesity[23]. Other studies have confirmed that the pandemic has been associated with reductions in routine assessments and treatment for various conditions[24] as well as in ophthalmology care contexts[5,6,27]”.
Additional reference: Chen, D.A,; Tran, A.Q.; Dinkin, M.J.; et al. Ophthalmic Virtual Visit Utilization and Patient Satisfaction During the COVID-19 Pandemic. Telemed J E Health 2022, 28(6):798-805 |

Reviewer 3 Report
Dear authors,
thank you for cunducting this study and writing the manuscript. It is well written and gives interesting insights to the impact of the pandemic of patient care. I just have some suggestions.
Why did you list smokers in the baseline characteristics? The data should either be discussed or left out as smoking does not seem to play a role in this study.
The study cohort is relatively small, consider going a little more into depth how this matters, especially when looking at subgroup formation.
The large majority of participants was female. Do you think this impacts your data? In which ways? Try adding a small section in the discussion.
There seems to be something missing in line 76.
Kind regards
Author Response
ijerph-1798250 |
|
The COVID-19 pandemic and ophthalmic care: a qualitative study of patients with neovascular age-related macular degeneration (nAMD) |
|
Reviewer comments |
Authors response |
Reviewer #3 |
Thank you for your comments and recommendations. We have addressed each point as follows: |
Why did you list smokers in the baseline characteristics? The data should either be discussed or left out as smoking does not seem to play a role in this study. |
Smoking was included in the Table as an important demographic descriptor and as it is a key risk factor for nAMD development and progression. We have however, amended the Table to remove this information. (Pg 3; Line 141)
|
The study cohort is relatively small, consider going a little more into depth how this matters, especially when looking at subgroup formation. |
Thank you for your comment. We agree that this is an important point. We are confident the sample is sufficient to provide data adequacy and have taken care to interpret findings based on the overall sample and not on particular sub-groups of patients.
|
The large majority of participants was female. Do you think this impacts your data? In which ways? Try adding a small section in the discussion. |
Thank you for your comment. We have addressed this by including an additional comment under the study limitations section (Pg 8; Line 268) by adding the following text: “The number of male participants recruited to the study was also relatively low”.
|
There seems to be something missing in line 76. |
Thank you. This sentence has been amended. |

Round 2
Reviewer 2 Report
The more relevant reference not included is: Rozon et al. Fear Associated with COVID-19 in Patients with Neovascular Age-Related Macular Degeneration.
Author Response
ijerph-1798250 |
|
The COVID-19 pandemic and ophthalmic care: a qualitative study of patients with neovascular age-related macular degeneration (nAMD) |
|
Reviewer comments |
Authors response |
Reviewer #2 |
|
The more relevant reference not included is: Rozon et al. Fear Associated with COVID-19 in Patients with Neovascular Age-Related Macular Degeneration |
Thank you for your suggestion. We have amended the text at line 260 accordingly.
“Findings from quantitative survey studies have also highlighted the possible influence of COVID-19 related fear on care continuity in ophthalmology care contexts. In one study[25], around half of participants with AMD or diabetic retinopathy were at least moderately concerned about vision loss due to missed or delayed treatment; and concern relating to COVID-19 exposure during appointments was a factor associated with higher loss to follow-up (odds ratio [OR], 3.9; 95% CI, 1.8-8.4). A further study[26] found a lower number of participants (16%) were fearful of visual loss due to difficulties in maintaining regular follow-ups. However, female participants were more likely to postpone appointments, potentially due to higher levels of COVID-19 related anxiety”.
Added references:
[25] Lindeke-Myers, A.; Zhao, P.Y.C.; Meyer, B.I. et al. Patient Perceptions of SARS-CoV-2 Exposure Risk and Association With Continuity of Ophthalmic Care. JAMA Ophthalmol 2021, 139(5):508-515. [26] Rozon, J.P.; Hébert, M.; Bourgault, S. et al. Fear Associated with COVID-19 in Patients with Neovascular Age-Related Macular Degeneration. Clin Ophthalmol 2021, 15:1153-1161.
|
